# Repetitive Transcranial Magnetic Stimulation for Major Depressive Disorder Comorbid with Huntington's Disease: A Case Report

Clémence Noiseux [1,†], Jean-Philippe Miron [1,*,†], Véronique Desbeaumes Jodoin [1], Tian Ren Chu [1], Sylvain Chouinard [2] and Paul Lespérance [1]

1   Department of Psychiatry, Centre Hospitalier de l'Université de Montréal, 1000 St-Denis, Montréal, QC H2X 3J4, Canada; clemence.noiseux@umontreal.ca (C.N.); veronique.desbeaumes.ccsmtl@ssss.gouv.qc.ca (V.D.J.); tian.ren.chu@umontreal.ca (T.R.C.); paul.lesperance.med@ssss.gouv.qc.ca (P.L.)
2   Department of Neurology, Centre Hospitalier de l'Université de Montréal, 1000 St-Denis, Montréal, QC H2X 3J4, Canada; sylvain.chouinard.med@ssss.gouv.qc.ca
*   Correspondence: jean-philippe.miron@umontreal.ca
†   Co-first authors.

**Abstract:** Huntington's disease (HD) is a rare genetic disorder resulting in progressive neurodegeneration leading to motor, cognitive and psychiatric symptoms. A high percentage of HD patients suffer from comorbid major depressive disorder (MDD). We are not aware of any literature on the use of repetitive transcranial magnetic stimulation (rTMS) for treating comorbid MDD in HD. We present the case of a 57-year-old man suffering from HD in which comorbid MDD was successfully treated with rTMS. Further work is required to better characterize the safety, tolerability and effectiveness of rTMS to treat comorbid MDD in HD.

**Keywords:** MDD; TMS; rTMS; neuromodulation; depression; chorea

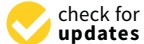



## 1. Introduction

Huntington's disease (HD) is a rare neurodegenerative condition usually caused by an inherited genetic mutation in the huntingtin gene. Neurodegeneration occurs in many brain regions, including the thalamus, cerebellum, hypothalamus, and basal ganglia [1]. The activity of various neurotransmitters is also altered in HD, including dopamine, glutamate and γ-aminobutyric acid (GABA) [2]. HD is mainly characterized by a hyperkinetic movement disorder known as chorea, resulting from basal ganglia degeneration. HD is also characterized by cognitive and psychiatric symptoms, with 15–69% of patients suffering from comorbid major depressive disorder (MDD) [3]. MDD is often one of the earliest presentations of HD [4], with patients at a significantly higher risk of suicidal behavior compared to the general population [5,6]. In this context, optimal treatment of MDD in HD patients is essential.

In HD patients suffering from comorbid MDD, antidepressants remain the first-line treatment [7]. Still, pharmacotherapy is burdened by limited effectiveness and several side-effects [8]. Given those limitations, new therapies have been developed and are now available for MDD. In the past decade, repetitive transcranial magnetic stimulation (rTMS) has emerged as a safe, well-tolerated and effective treatment for treatment-resistant depression [9]. rTMS uses a rapidly alternating and powerful magnetic field to modulate neural activity in specific brain regions known to be abnormal in MDD. The most recent literature indicates that rTMS consistently achieves response and remission rates of 40–55% and 25–35% in treatment-resistant depression, respectively, with little to no side-effects [7].

There is a limited number of publications reporting the use of rTMS for treating comorbid MDD in neurological disorders. A meta-analysis including seven studies reported

that rTMS effectively reduced depressive symptoms in Parkinson's Disease [10]. A recent double-blind, sham-controlled study reported significant improvement in the depressive symptoms of eleven patient suffering from post-stroke depression with high-frequency rTMS [11]. So far, we are not aware of any reports on the use of rTMS in HD patients suffering from comorbid MDD.

## 2. Case Report

We report on the case of a 57-year-old man suffering from both HD and MDD. The patient initially had positive HD genetic testing in 1993, which was conducted given his positive family history for the disease. Initial MDD symptoms appeared in the early 2000s and diagnosis of HD was made in 2005. The patient was initially started on venlafaxine, to which mirtazapine was added 6 months later. The episodic increase in symptoms was managed by medication dose increase. The patient initially responded favorably to antidepressant medication, achieving stability for several years with only few relapses. Around 2010, bupropion was added to the regimen because of an increase in depressive symptoms but was not tolerated and therefore discontinued. Around 2015, valproic acid was introduced to manage impulsivity issues. Depressive relapses became more severe in 2018, and medication changes did not bring any additional benefits. rTMS was therefore offered in early 2019.

Treatment consisted of a high-frequency (HF) 20 Hz rTMS protocol (5 s ON, 25 s OFF, 3000 pulses, ~15 min duration) delivered over the left dorsolateral prefrontal cortex (DLPFC) using the BeamF3 algorithm [12] at 120% of the resting motor threshold (rMT), using R30/X100 stimulators equipped with a DB80 coil (MagVenture, Farum, Denmark). We used the same intensive protocol reported in a previous publication from our group for which we observed good results with older patients [13]. High-frequency protocols seem to have a higher efficacy in older adults with cortical atrophy [14,15]. The DB80 coil was chosen given the light diffuse cerebral atrophy seen on MRI, and a quite elevated motor threshold with a B-70 coil. The treatment course consisted of three daily sessions every weekday over 2 weeks, for a total of 30 sessions. The movements induced by the chorea was initially an issue, as the patient was involuntarily moving away from the coil at times. This was addressed by having him lie on his back instead of sitting during treatment, securing the head between the coil and the headrest (Figure 1). The patient did not report any pain or side effects during treatment.

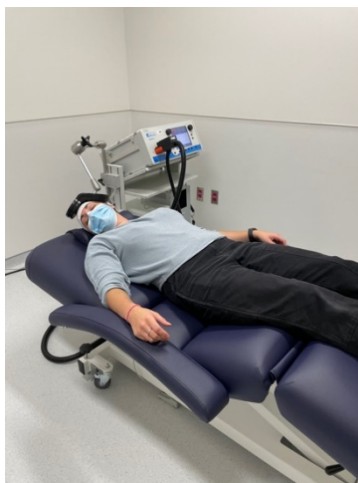

**Figure 1.** Supine treatment position to limit chorea.

Symptoms were assessed pre- and post-treatment using the Montgomery–Asberg Depression Rating Scale (MADRS). Remission was defined as a score < 10. After his first treatment series in 2019, he achieved remission, with MADRS scores going from 24 at baseline (moderate depression) down to 5 (normal range) at follow-up. Following rTMS,

the patient reported increased energy as well as an increased interest in his daily activities. He remained stable until 21 months later without any maintenance treatment, after which he experienced a depressive recurrence. The patient subsequently underwent a second course of treatment using the same protocol, after which he once again reached remission, with MADRS scores decreasing from 18 at baseline (mild depression) to 10 post-treatment (mild depression). To optimize response, the patient underwent maintenance treatments once a week over 7 weeks (three sessions on weeks 1 and 2, two sessions on weeks 3, 4 and 5, and one session on weeks 6 and 7). His MADRS score went down to 6 (normal range) at follow-up.

## 3. Discussion

Several hypotheses have been put forward to explain the high comorbidity between HD and MDD. One explanation is that depressive symptoms are the direct result of cerebral degeneration (more specifically neuronal loss in the striatum), also affecting limbic structures [16]. This in turn can affect prefrontal activity, a region central to MDD [17]. Dopaminergic activity is also known to be altered in HD. Even though dopamine (DA) activity is increased during the early stages of HD, DA deficit leading to hypokinesia is the hallmark of the later stages of the disease. In and of itself, alterations in DA neurotransmission could contribute to depressive symptoms, while also affecting glutamate activity, leading to excitotoxicity [18].

Multiple studies have attempted to characterize the antidepressant effects of rTMS in MDD. The effects of rTMS on cell-signaling pathways, including an increase in cell proliferation and BDNF in limbic areas, could be part of its antidepressant effect [19,20]. Left-DLPFC rTMS has also been shown to modulate DA activity in striatal areas and beyond, such as the anterior cingulate cortex and orbitofrontal areas, which have a central role in MDD [21–23].

The effect of rTMS on chorea was not formally monitored during treatment, and neither the patient nor the medical team noticed any changes on that front. The effect of rTMS applied over the motor cortex in the context of motor disturbances in patients with HD has been found to be highly variable [24]. Some studies reported a decrease in movement disorders in HD patients [25], while others reported no significant changes [26]. Still, as rTMS is known to increase DA activity in the striatum [27], it could potentially influence chorea. Future studies should prospectively assess motor symptoms using validated scales such as the Unified Huntington Disease Rating Scale (UHDRS) [28].

Significant brain atrophy is often observed in patients with HD. The presence of regional brain atrophy might attenuate rTMS current density in the cortex, but there is evidence that rTMS can have clinical benefits in populations with known brain atrophy, such as in Alzheimer's disease [29,30]. Therefore, rTMS should not be excluded in HD patients suffering from comorbid MDD simply because of brain atrophy.

## 4. Conclusions

We report on the first demonstration of safe and successful treatment of MDD in a HD patient with rTMS. Of note, initial rTMS-induced antidepressant effects persisted almost 2 years post-treatment in our patient. Similar mood improvements were observed during both courses of rTMS treatment. Chorea might be a barrier to rTMS therapy in HD patients suffering from comorbid MDD, jeopardizing reliable coil placement during stimulation. Instead of the classical seating position, we therefore recommend a supine position, decreasing the odds of movement away from the optimal stimulation area. As this study is a case report, we cannot generalize the results and future studies need to replicate these findings in larger samples and should also assess whether rTMS influences chorea and cognition in HD patients suffering from comorbid MDD.

**Author Contributions:** C.N.: writing original draft; J.-P.M.: conceptualization, writing—original draft, supervision, writing—review & editing; V.D.J. and T.R.C.: conceptualization, supervision, writing—review & editing; S.C.: writing—review & editing; P.L.: writing—review & editing. All authors have approved the final article. All authors have read and agreed to the published version of the manuscript.

**Funding:** This research received no external funding.

**Institutional Review Board Statement:** Not applicable.

**Informed Consent Statement:** Written informed consent has been obtained from the patient to publish this paper.

**Acknowledgments:** The authors are pleased to acknowledge Sylvie Tieu and Ana Baker, clinical nurses, who provided treatments to the patient.

**Conflicts of Interest:** The authors report no potential conflict of interest.

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
