# Peer review of "Repetitive Transcranial Magnetic Stimulation for Major Depressive Disorder Comorbid with Huntington’s Disease: A Case Report"

_neurosci, doi:10.3390/neurosci2040029_

Round 1

Reviewer 1 Report

The case report presented by Clémence Noiseux et al., is interesting. The manuscript is well written and structured. However I would suggest the authors to improve the introduction, in particular they should mention previous studies that used the stimulation, in patients affected by other diseases. At the same time the authors should explain also the limitless of this approach in the discussion of the manuscript.

Author Response

Thank you for your constructive comments. We have now improved the introduction section with rTMS studies regarding other neurological diseases with comorbid major depressive disorder. We have also improved the methodology and the limitation section of the manuscript.

Reviewer 2 Report

This is a very interesting case report. This a unique way of treating MDD especially in patients with coexisting other neurological degenerative  conditions. This patient was resistant to standard pharmacological treatment and responded well. TMS is a growing concept. 

Has there any standardization of the TMS dosing schedule? How did the authors come up this schedule? Please specify if possible. 

Otherwise, I really liked this article. Thank you for your hard work.  

Author Response

Thank you for your constructive comments. There is no standardization for dosing TMS with cortical atrophy. Few studies showed that high-frequency protocols seem to have a higher efficacy in older adults with cortical atrophy 1-2. Moreover, a prospective, randomized, sham-controlled, double-blind study using a higher stimulation intensity, a larger number of pulses, and a larger number of sessions showed the efficacy of rTMS in elderly patients with vascular depression 3.  We decided to use an intensive protocol that we reported in a previous publication from our group for which we had good results with older patients 4. We have now added this information to the manuscript.

References

(1) Ahmed, M. A.; Darwish, E. S.; Khedr, E. M.; serogy, Y. M. E.; Ali, A. M. Effects of Low versus High Frequencies of Repetitive Transcranial Magnetic Stimulation on Cognitive Function and Cortical Excitability in Alzheimer’s Dementia. Journal of Neurology 2012, 259 (1), 83–92. https://doi.org/10.1007/s00415-011-6128-4.

(2) Guse, B.; Falkai, P.; Wobrock, T. Cognitive Effects of High-Frequency Repetitive Transcranial Magnetic Stimulation: A Systematic Review. Journal of Neural Transmission 2009, 117 (1), 105–122. https://doi.org/10.1007/s00702-009-0333-7.

(3) Jorge, R. E.; Moser, D. J.; Acion, L.; Robinson, R. G. Treatment of Vascular Depression Using Repetitive Transcranial Magnetic Stimulation. Archives of general psychiatry 2008, 65 (3), 268–276. https://doi.org/10.1001/archgenpsychiatry.2007.45.

(4) Jodoin, V. D.; Miron, J.-P.; Lespérance, P. Safety and Efficacy of Accelerated Repetitive Transcranial Magnetic Stimulation Protocol in Elderly Depressed Unipolar and Bipolar Patients. Am J Geriatric Psychiatry2019, 27 (5), 548–558. https://doi.org/10.1016/j.jagp.2018.10.019.

Reviewer 3 Report

I find the content of the manuscript to be exceedingly interesting.  HD is a terrible disease; the research described ever so lucidly in the manuscript promises to provide a useful treatment particularly so if the results of an expanded study leads to similarly excellent outcomes.

I enthusiastically recommend that the manuscript be accepted for publication.

Author Response

Thank you for your kind comments.

Round 2

Reviewer 1 Report

The authors satisfied all my concerns. I would accept the manuscript in the present form for publication.